# Influence of Rock Dust Additives as Fine Aggregate Replacement on Properties of Cement Composites—A Review

**DOI:** 10.3390/ma15082947

**Published:** 2022-04-18

**Authors:** Magdalena Dobiszewska, Orlando Bagcal, Ahmet Beycioğlu, Dimitrios Goulias, Fuat Köksal, Maciej Niedostatkiewicz, Hüsamettin Ürünveren

**Affiliations:** 1Faculty of Civil and Environmental Engineering and Architecture, Bydgoszcz University of Science and Technology, 85-796 Bydgoszcz, Poland; 2School of Engineering, Construction Science and Management, Tarleton State University, Stephenville, TX 76402, USA; bagcal@tarleton.edu; 3Department of Civil Engineering, Adana Alparslan Türkeş Science and Technology University, Adana 01250, Turkey; abeycioglu@gmail.com (A.B.); hurunveren@atu.edu.tr (H.Ü.); 4Department of Civil and Environmental Engineering, University of Maryland, College Park, MD 20742, USA; dgoulias@umd.edu; 5Department of Civil Engineering, Yozgat Bozot University, Yozgat 66900, Turkey; fuat.koksal@yobu.edu.tr; 6Faculty of Civil and Environmental Engineering, Gdańsk University of Technology, 80-233 Gdańsk, Poland; maciej.niedostatkiewicz@pg.edu.pl

**Keywords:** concrete, mortar, waste management, rock dust, concrete strength, concrete durability

## Abstract

Concrete production consumes enormous amounts of fossil fuels, raw materials, and is energy intensive. Therefore, scientific research is being conducted worldwide regarding the possibility of using by-products in the production of concrete. The objective is not only to identify substitutes for cement clinker, but also to identify materials that can be used as aggregate in mortar and concrete productions. Among the potential alternative materials that can be used in cement composite production is rock dust of different geological origin. However, some adversarial effects may be encountered when using rock dust regarding the properties and durability of mortars and concrete. Therefore, comprehensive research is needed to evaluate the adequacy of rock dust use in cementitious composite production. This paper presents a comprehensive review of the scientific findings from past studies concerning the use of various geological origins of rock dust in the production of mortars and concrete. The influence of rock dust as a replacement of fine aggregates on cementitious composites was analyzed and evaluated. In this assessment and review, fresh concrete and mortar properties, i.e., workability, segregation, and bleeding, mechanical properties, and the durability of hardened concrete and mortar were considered.

## 1. Introduction

The industrialization and advancement of society generate large quantities of waste, creating a significant impact on the environment. Worldwide, waste generation has increased greatly in recent years and shows no signs of slowing down, other than the temporary effects of the COVID19 pandemic in construction. Each year, approximately 2.01 billion tons of municipal solid waste are generated worldwide, at least 33% of which is not well managed in terms of being environmentally friendly [1]. This means that the daily waste generation per capita worldwide ranges widely from 0.11 and 4.54 kg, with an average of 0.74 kg [1]. By 2050, the amount of municipal solid waste generated worldwide is expected to increase by approximately 70% and reach 3.4 billion metric tons. Less than 20% of waste is recycled each year, with vast amounts still sent to hazardous open landfill sites, thus posing a significant threat to the environment [1]. Coal combustion residues, such as blast furnace slag and rock dust of different geological origins and solid wastes produced in various industrial processes and mining sectors, constitute a waste group of their own. These industrial and mining wastes are complex and complicate the task of safe disposal and/or environmentally sound use in terms of quality and quantity. In most industrialized countries, waste disposal has become a cause for concern due to limited site conditions and stringent environmental standards. Thus, there is a pressing demand for authorities and agencies to ensure sufficient waste treatment and disposal services in order to attain better efficiency in waste management, particularly focusing on the reuse of waste materials.

Industrial and economic development has intensified activities in the construction industry, demanding an increase in the production of building materials. It is undeniable that concrete is the most extensively and generally used construction material worldwide. The continuous rapid growth of urban areas and infrastructure has led to an increase in demand, and this puts excessive pressure on the concrete industry for the production of large quantities of concrete to meet that demand. According to data compiled by the U.S. Geological Survey in 2019, the yearly global production of Portland cement was about 4.1 billion mt and it is forecasted to increase to 5.8 billion mt in 2050 [2,3,4]. Assuming that an average of 350 kg of cement is used per cubic meter of ordinary concrete, it can be estimated that the annual production of concrete in the world amounts to about 12 billion mt, which leads to an annual global average consumption rate of about 1.6 tons of concrete per person. The production of such an amount of concrete also requires 9 billion mt of aggregates and 2.2 billion mt of fresh water. Environmental constraints considerably decrease the scale of the natural deposits, which are used for the manufacture of cement and natural aggregates. The use of approximately 40% of the world’s resources, such as water, fine and coarse aggregates, and wood, is the responsibility of the construction industry [5]. Not only over-exploitation and finite natural resources, but also the growing increase in the amount of various industrial waste and the lack of storage and landfill space have led to the development of extensive research throughout the years for assessing the potential use of these wastes in building materials production. Sustainable development principles, which is expressed as the rational management of non-renewable resources and the substituted use of these resources with recycling wastes, are compatible with such efforts.

There is a sort of material that can be utilized not only as a substitute for cement clinker, but also as a substitute for natural raw materials for the production of building materials [6,7,8,9]. In conjunction with the rapid increase in concrete production, the demands for natural aggregates are also increasing. To meet the increasing demand for aggregates, natural river sand, regarded as the most appropriate and commonly used fine aggregate in the production of mortar and concrete, is comprehensively exploited. This has led towards the uncontrollable exploitation of natural aggregate and serious environmental and economic concerns [10,11,12]. The mining of river sand has a very harmful influence on the environment, such as river flow, erosion levels, and aquatic habitats. Dredging a riverbed can destroy not only riverbanks, but also the habitat occupied by the bottom-dwelling organisms. The water may become cloudy due to the sediment that will form during the dredging operations, the fish may drown due to the sediment that will form, and the sunlight from which the aquatic vegetation feeds may be blocked [13,14,15,16]. Due to the massive depletion of river sand and strict environmental requirements, there is a shortage of sand for building materials production in many countries around the world [15,17,18,19,20,21]. Furthermore, there is currently a deficiency of good quality natural sand that may be used in concrete production in many regions of the world [15,17,18,22]. In some countries, preventive restrictions on the extraction of river sand have been introduced in order to protect valuable natural areas [13,14,15,16]. Considering that natural sand is about 35% of the concrete volume, combined with the increased demand from construction, this implies a serious shortage. Such shortage creates challenges for the concrete industry to identify alternative solutions. As mentioned earlier, rock dust may be a promising alternative for fine aggregate in mortars and concrete. These inert fillers, which may be composed of rock of different geological origin from the grinding process, can be used to enhance both particle size distribution and packing density. The optimalization of the cementitious and aggregate blended systems has become the art of maximizing the use of various by-products and their positive synergetic effects [23].

The following sections of the manuscript present and synthesize the findings of various studies when rock dust is to be used as a replacement of fine aggregate in mortars and concrete.

## 2. Rock Dust Characteristics

Aggregate is defined by the European standard EN 12620 [24] as granular materials of natural, manufactured, and recycled origin used in the construction industry. Similar definitions are provided from various ASTM standards [25,26]. The aggregate grains are originally part of the parent rock and are divided into fine fractions by either natural factors (i.e., by weathering and abrasion) or artificially by mechanical grinding and crushing of the rock. Thus, many properties of aggregates, namely chemical and mineral composition, petrographic characteristics, density, water absorption, and strength, are dependent on the bedrock. On the other hand, other characteristics of the aggregates, such as shape, grain size, and surface texture, depend primarily on the technique used to crush the bedrock. The name natural aggregate covers all mineral aggregates that come from deposits, i.e., gravel and sand (fine aggregate) and pebbles obtained from loose rock materials, as well as crushed aggregates produced from mechanically treated rocks.

Approximately 4 billion mt of aggregates are produced and consumed in Europe and almost 91% of these aggregates are obtained from natural deposits [27]. In the US, the estimated annual output of construction aggregates produced for consumption was around 2.5 billion mt in 2020, with an estimated increase of 3–5% per year [2,28]. Crushed aggregates are mainly produced from igneous (basalt, melaphyre, diabase, porphyry, gabbro, and granite), metamorphic (amphibolite, gneiss, serpentinite), and sedimentary rocks (limestones, dolomites, sandstones, greywackes,). Large amounts of waste material in the form of rock dust are generated during the extraction and mechanical treatment of rocks, and then as a result of their categorization. Rock dust is also obtained during the aggregate production process for asphalt mixes. Waste dust accounts for around 5% of the aggregate mass used in the production of asphalt mixes. This means about 5000 tons of waste dust are produced annually in an average size asphalt mixture plant. Similar dusty waste is generated in dimensional stone factories, where mainly granite and marble are processed. They are used for paving stones, floors, cladding panels, tombstones, monuments, and statues. About 68 million tons of rock are processed annually in the stone industry around the world [29]. Countries where over a million tonnes of stone are processed annually include Italy, Portugal, Greece, France, Turkey, USA, Brazil, South African, India, and China [29]. When cutting, grinding, and polishing the rock blocks, water is used to cool and moisten the saws and polishing equipment. As a result of such a processing, semi-liquid sludge is formed as a waste in the amount of about 20–30% [30,31,32,33,34,35,36]. This waste is collected in settling tanks and then stored in the pulp form in landfills [18]. Part of the water contained in the pulp penetrates into the ground and paves the way for fine dust particles, which fill the voids and gaps in the ground. This causes significant soil permeability reduction, which negatively affects the soil fertility and groundwater level [18,37,38]. Part of the water is evaporated, and then dried dust is carried by the wind to the atmosphere, posing a threat to people and the environment [29,39].

According to the European standard EN 12620 [24], natural mineral dust is a fraction of aggregates with grain sizes smaller than 0.063 mm. Dust in crushed aggregate is generated from the crushing of the bedrock. On the other hand, uncrushed natural aggregate may contain dust resulting from natural weathering processes, as well as clays. These dusts can coat the surface of aggregate grains, which reduces the adhesion of the cement paste to the aggregate grains, resulting in a decrease of concrete strength. Moreover, clay grains may be adsorbed on the cement grain surface and create a water-impermeable coating, which delays hydration [29]. Additionally, due to the propensity of clay minerals to swelling due to the presence of water, the volume stability of mortars and concretes is influenced. Therefore, clay grains are not desirable as aggregate in concrete. Similar definitions and guidelines are provided by the American Concrete Institute, ACI, and the American Society for Testing and Materials, ASTM, for aggregates to be used in concrete, and in terms of classifications that are based on bulk density (i.e., unit weight) [25], mineralogical composition [40], and particle shape [26]. The National Stone, Sand, and Gravel Association, NSSGA [41], provides further guidance on the characteristics, physical properties, and mineralogical composition of rock dust.

Standards adopted by agencies in various countries contain guidelines regarding the limit content of dust in aggregate to be used in the production of concrete. The European standard, EN 12620 [24], presents the categories of maximum dust contents to be used by aggregate producers. The total dust content in the fine aggregate is considered to be harmless if it less than 3% by weight of the aggregate. The content of grains smaller than 75 µm in coarse aggregate cannot exceed 4%, according to the British standard BS-EN 12620 [42], whereas the content of fine aggregate depends on the concrete application and may not exceed 10–14%. On the other hand, the American guidelines specified in ASTM C33 [25] limit the maximum content of dust to 3% in the aggregate used for the production of concrete exposed to abrasion and 5% for other concretes. The aggregate must be washed before used in concrete in the case of excess on the permissible dust content. The fine-grained material obtained in this way is a dusty waste. It can be pointed out that the classification of dust as a deleterious additive in concrete solely on the basis of its grain size is incorrect [13,29]. As indicated earlier, mineral dusts that adhere to the surface of aggregate particles are not desirable in concrete. However, as shown later in this paper, when they are added to concrete or mortar, they can be beneficial in terms of the properties of hardened composites [18,33,43,44,45,46,47,48,49,50].

The characteristics of mineral dusts presented in this review were limited to dusts from limestone, marble, granite, and basalt rocks, which represent the most rock dust in concrete production and thus examined on past studies. Marble is formed by the metamorphism of limestone and dolomite over a wide range of pressures and temperatures. In the process of grain recrystallization, carbonate sedimentary rocks are transformed into crystalline rocks. Thus, petrographically, marble is a limestone. However, in published studies on the use of rock dust in cement composites, a distinction is made between limestone dust and marble dust. Therefore, in this paper, it was also decided to keep such a division. The oxide composition of rock dusts is presented in Table 1.

Natural rock dust from rock fragmentation has a rough surface, sharp edges, and irregular shapes. Examples of the texture of limestone, marble, and basalt rock dusts observed under a scanning electron microscope are shown in Figure 1.

A question arises whether, in light of the applicable standards, rock dust can be used and classified as additive for mortars and concrete. In the case of fillers or fine aggregates, their suitability for use in concrete is determined on the basis of EN 12620 [24] and EN 13055-1 [58] standards. Rock dust can therefore be considered as type I concrete additive, i.e., chemically inert mineral fillers. According to the EN 12620 [24] standard, the filler is the aggregates, the majority of which pass through a sieve of 0.063 mm and provide specific features by their addition to the construction materials. However, the European standards do not explicitly specify the permissible content of filler aggregates in mortars and concrete mixes. Nevertheless, concrete specification should provide the type and amount of this additive and the rock dust origin.

Thus, it should be stated that, considering the applicable standards, the addition of dusts as mineral fillers in the composition of mortars and concretes is determined by these cement composites properties, which may not be affected by the addition of dusts.

## 3. Fresh Concrete and Mortar Properties

The use of stone dusts as filler in concrete has a significant effect on workability. Fine filler additives in appropriate quantity improve the workability of cementitious materials and may not increase the water requirement [59,60]. However, for a constant w/c ratio, when too much powder content is used, more water is necessary in order to wet the grains surface, resulting in reduced mixing water and consequently poor workability [54,57,61]. On the other hand, crusher dust consumes more water than sand because of its rough texture. Thus, it causes a reduction in workability [62]. Hameed and Sekar [63] stated that 50% replacement of marble dust with river sand improves the workability of mortar. Janakiram and Murahari [64] investigated the workability of concrete where quarry dust and marble dust were used in various proportions of 0%, 10%, 20%, 30%, 40%, 50%, and 60% instead of natural sand. They reported that the workability decreased for all replacement percentages and a reduction of 28.57% was obtained for the 60% replacement. Idrees and Faiz [65] used marble powder and quarry dust in concrete as a replacement of sand at the percentages of 12.5% and 25% separately, and 25%, and 50% as combined replacement in equal proportions. They reported that marble powder negatively affects the workability while quarry dust improves it. They also indicated that the use of quarry dust increased slump while marble powder reduced it. It was concluded that the combined replacement of marble powder and quarry dust moderately improved the workability of concrete. Other authors also observed that adding marble powder into concrete or mortar shows a reduction in workability [30,38,66,67,68]. Several studies reported in the literature examined the influence of limestone powders in terms of concrete workability [69,70,71,72,73,74]. Some studies concluded that limestone powder decreases workability of concrete [70,72,74], while others reported improvements in workability [69]. Filler, dilution, and morphological effects of limestone powder play a role on the flowability of concrete [71]. Dobiszewska et al. [56] analysed the workability of concrete by replacing 5%, 10%, 20%, and 30% of sand (by mass) with basalt powder. It was concluded that workability decreased because of the much greater specific surface area of the basalt powder in comparison to the sand.

Segregation and bleeding in cement-based materials are two effects related to the loss of homogeneity. Segregation is observed as the settlement of aggregates within the mortars and concrete. Bleeding is associated with excess water rising to the surface of a highly fluid concrete mixture. Bleeding and segregation can be controlled by using well graded aggregates, finer cement, proper water to cement ratio, entraining agents, and mineral additives [75]. Uniform mixing is also important in reducing the propensity to bleeding and segregation. The use of fine granulated materials reduces bleeding and segregation by creating a longer path for water to rise to the surface, blocking the pores and improving the cohesion of the mix [59,75,76]. There are many studies on the effects of quarry dust, such as marble dust, granite dust, crushed rock dust, and limestone powder, on the bleeding and segregation of cement-based composites [77,78,79,80,81]. In all the studies examined, it was emphasized that the use of non-pozzolanic fillers in mortar or concrete mixtures increased bleeding and segregation resistance. Danish and Mohan Ganesh [78] reported that a reduction of 65.2% in the bleeding resistance of self-compacting concrete (SCC) was obtained by using marble powder. Schankoski et al. [82] indicated that no bleeding occurred in the quarry dust pastes which had a lower viscosity than those with limestone fillers. It was mentioned that bleeding is prevented due to the higher surface area and longer shaped quarry dust particles [82]. Elyamany et al. [79] conducted a study on the effects of various pozzolanic, such as silica fume and metakaolin, and non-pozzolanic fillers, such as limestone powder, granite dust, and marble dust, on the bleeding and segregation of self-compacting concrete. It was concluded that marble and granite powders showed better bleeding resistance compared to other filler types used. It was also concluded that a significant effect on bleeding was observed with a filler content of 15.0%. Nguyen et al. [80] emphasized that SCC would have sufficient bleeding resistance if 30% of dolomite powder was replaced with pozzolanic fillers.

Further, air content is a very important ingredient for cementitious materials because it directly affects the mechanical and durability properties cementitious materials. Cementitious based composites contain two types of air, namely entrapped and entrained. Entrapped air occurs naturally in the mix during mixing operations. These voids are convoluted and interrelated. On the other hand, entrained air is formed by the addition of air entrained admixture into the mixture. Those voids are spherical in form and independent from each other. In a conventional concrete (non-air entrained concrete) with a suitable mixture and sufficient compaction, the air content is around 1.5–2%. Air content can be increased up to 4–8% by using air entrained admixture for the improvement of freeze-thaw resistance for cold weather concreting [59,75]. The factors affecting the air content of concrete can be listed as follows: water and cement contents, maximum size and grading of aggregate, mixing and compaction of concrete, temperature of concrete, admixtures (mineral and chemical), and the use of fillers (stone powder, rock dust etc.). As a replacement of sand, rock powders and quarry dusts are the most preferable filler materials as ultrafine aggregates filling the voids to control or decrease the air content of cementitious materials. The use of very fine materials, with larger specific surface area than cement and in adequate quantities, reduce the air content of concrete [76,83,84,85]. However, the use of excessive rock dust particles in relation to the voids between cement and sand particles has a reducing effect on pore filling, resulting in an increase in air content due to a reduction in packing density [86,87,88,89]. Therefore, the optimum substitution of fine rock powder into cementitious composites is an important consideration.

It is seen that there are conflicting interpretations in the literature on workability, but in general, the use of fillers affects the workability negatively by effectively changing the water/cement in the concrete. For this reason, when stone dust is used, preliminary tests must be performed, and water/cement adjustment must take into account the stone dust used instead of the fixed water-cement ratio. In addition, it is seen that there is a consensus in the literature that the effect of stone dust on segregation and bleeding resistance is very significant. It is also seen that the use of stone dust reduces the air content by filling the voids in the concrete. Although this situation seems positive in terms of reducing the permeability of concrete, it should also be investigated in terms of durability problems, such as freezing and thawing.

## 4. Hardened Mortar and Concrete Properties

### 4.1. Compressive Strength

The introduction of stone powder, which partially replaces fine aggregate, affects the properties of mortar and concrete physically and mechanically. The relative compressive strength of mortar and concrete with rock powder addition at a different curing age are shown in Figure 2, Figure 3, Figure 4, Figure 5, Figure 6 and Figure 7. Soroka and Stern [90] noticed that the addition of rock dust powder positively affects the cement mortars mechanical properties. They observed that as the amount of dust used sand replacement and the fineness of the dust increases, so does the strength of the mortar. Similar conclusions were obtained with studies examining the usage of rock dust as a partial replacement for sand in mortars and concrete: limestone dust [44,46,51,55,91,92], marble dust [12,30,33,34,61,67,76,86,92,93,94], granite dust [14,17,18,43,54,95], and basalt powder [46,55,56,57,96].

The filler role of stone powder is primarily responsible for the improvement in strength in cement composites with rock dust addition. As mentioned earlier, the process of the heteronucleation of cement clinker hydrates on dust particles mechanically improves the cement matrix microstructure and interfacial transition zone [12,30,46,56,67,99,112,113,114,115]. The chemical composition of the parent rock and the rock type from which the powder comes has a minor impact on the rock dust operation mechanism in this situation [46,116]. Much more important and dominant influence is the fineness of the rock dust. Nevertheless, it should be noticed that the rock dust specific surface area affects to a greater extent the mortar and concrete mechanical properties when rock dust is used as a partial cement substitution rather than as fine aggregate replacement. As observed earlier, cement substitution with inert additives of finer particles size and greater fineness compared to cement particles results in the increase of hydration products nucleation sites. Hydration products crystallize on cement particles as well as on the rock powder surface, which contributes to an increase of the rate and hydration degree of cement clinker. This leads to an increase in the content of C-S-H phase, decrease of cement paste porosity, and therefore to an increase in cement matrix strength, particularly in the early hydration process [117]. When cement is replaced with rock dust of larger particles diameter than cement grains, it results in a reduction of the specific surface area where hydrates can crystallize. This action leads to lower hydration rate and degree of hydration and lower early strength at early ages. The addition of stone dust as sand substitution does not affect cement content, and thus the nucleation centers specific surface area increases in each case. Therefore, in this situation, it is irrelevant if the fineness of stone dust is larger or smaller than cement.

Rock dust is an inert filler and thus contributes in filling a greater range of the intergranular free space in cement composites. This results in the densification of the cement matrix, which leads to lower porosity and therefore higher strength [12,95,102,103,106,118]. Roy et al. [119] discussed the particular role of particle packing in achieving optimal mortar and concrete properties. The more regular cement grains dispersion, and therefore the faster hydration of the clinker phases of cement, occurs by the addition of microfiller grain [54,118,120].

Uchikawa et. al. [98] indicated that the increase in the strength of concrete by substitution of fine aggregate with rock dust is achieved by the increase in the density of the hardened concrete structure due to the generation of the C-S-H phase during the pozzolanic reaction, in addition to the mineral powder’s filling property. Abdelaziz et al. [55] observed the same in which compressive strength of mortar increases with basalt dust addition. Such an effect was attributed to the filler effects as well as to basalt pozzolanic activity. The reaction result of active silica and alumina ions in basalt with the calcium hydroxide CH in the cement pore solution, an additional amount of C-S-H phase is formed. In turn, this results in the increase of cementitious matrix density and strength improvement. Other researchers reached similar conclusions in regard to the slightly greater strength of mortars when basalt powder additive was used compared to mortars with other rock dust [46,55,121].

In the case of limestone and marble powder, the development in concrete’s compressive strength is connected with the physical and chemical effects of dust. The dominant reason is due to the physical filler effect of mineral powder. This results in filling the spaces between the cement grains and refining the pore structure, enhancing the concrete matrix microstructure, and thereby a strength increase. The chemical effect of limestone and marble powder concerns the reaction of calcium carbonate CaCO_3_ and alite C3A available in the cement [53] and results in the formation of calcium carboaluminate hydrates [63,92]. This increases the degree of hydration reactions and reduces porosity especially at early ages of hydration [122,123,124]. Thus, it contributes to an increase in early age strength [125,126].

However, with a certain rock dust content replacing fine aggregate, decreases in the mortar and concrete strength were observed [12,61,63,92,97,107,127,128]. Fine dust particles feature a large surface area and therefore need more water for moistening the grain surface. However, when the w/c ratio is kept constant, the increase in dust content leads to the reduction of available water necessary for hydration of cement clinker phases, poor workability, and thus poor compactness and a decrease in compressive strength [54,57,61]. Alyamac and Aydin [94] observed that high dust content leads to an improper grain-size distribution. This results in larger free space between particles and therefore strength reduction. When rock dust particles are in excess of the voids between cement and sand particles, then the particles push each other apart, leading to a reduction in packing density, and thus a reduction in compressive strength [86,88,89]. Hence, the pore filling effect is being downplayed. Once the optimum substitution level is reached, any higher amount of rock dust increases the surface area of particles instead of filling up the voids. The increase in the surface area requires an excess amount of cement to bind dust particles as well as aggregate grains. When the cement content is constant, a strength reduction is observed at higher rock powder inclusion [107]. Additionally, the presence of excessive permeable voids accelerates crack propagation and crack connectivity, thus, resulting in strength reduction at higher substitution rates of sand with rock powder [86,106]. Knop et al. [117] confirmed that high amount of very fine dust particles leads to the agglomeration as a consequence of the inter-particle interaction which generated massive grain aggregate formation with a diameter exceeding even 100 µm. As a result, it decreases the effective specific surface area and causes lower particle packing density, which directly affects strength reduction.

In general, the addition of rock dust has a positive effect on the strength as it fills the concrete voids and reduces the porosity. In addition, when the water/cement ratio is not modified according to the added stone powder, since the water required for cement hydration is used by rock dust, it can significantly reduce the compressive strength as well as the workability of the concrete.

### 4.2. Tensile and Flexural Strength

Few studies have addressed the impact on tensile and flexural strength of cement composites with rock dust addition as a fine aggregate. Overall, an increase in the rock powder replacing fine aggregate, an increase in tensile and flexural strength were observed [30,36,86,89,92,129,130,131]. As in the case of compressive strength, the increase in tensile and flexural strength is related mainly by the filler action of fine rock dust particles. The fine rock dust particles fill the voids in the cement matrix, and therefore a denser microstructure contributes to an increase in strength properties [89,132]. Some studies pointed out that the effects of rough surface texture and irregular shape of rock dust particles are the most significant parameters in increasing flexural and tensile strength of cement-based materials [133]. Such rock powder particle properties might improve the adherence of the aggregate phase to the cement paste, resulting in better bonding on the crack route created throughout the split tensile and flexural strength testing. This enhances the strength properties [12,92,106,107]. The decrease in porosity and improvement of the strength of both the cement paste matrix and the interfacial transition zone might be ascribed to the improvement in bond strength [115,134,135].

However, above a certain amount of stone dust replacing sand (i.e., about 30%), a decrease in h is observed. The increased fine content may also increase the pores in the concrete, which explains the flexural strength reduction [103,136].

Results similar to compressive strength in tensile and flexural strength appear in the literature, but the most important issue here is the rough surface texture and irregular shape of rock dust used.

## 5. Concrete and Mortar Durability

Galetakis and Soultana [39], as well as many other authors, have asserted that permeability is one of the most important factors characterizing the durability of concrete. The permeability of concrete is often measured based on its resistance to allow the penetration and movement of aggressive substances within its mass. The published research results indicate that concrete with the addition of different mineralogical origin rock waste demonstrated lower water permeability as compared with conventional concrete [33,35,92,137,138]. A study on the use of marble waste as coarse aggregate replacement conducted by Ulubeyli et al. [139] found that marble waste acted as a filler, reducing the gaps within the hardened concrete, thus providing a less porous structure of concrete. It can be stated that water permeability depends primarily on the capillary pores volume. However, Kurdowski [118] concluded that permeability is determined not only by the total porosity, but also to the distribution, tortuosity, shape of pores, as well as their size and continuity. The study conducted by Holly et al. [140] supported this concept by demonstrating a remarkable impact of the interconnectivity of cement paste pores and the pore size distribution on permeability. Menadi [22] observed a reduction in the water penetration depth with an increase in limestone powder content. This is the effect of the improvement of pore structure in the interfacial transition zone. The increase in concrete water permeability with the increase in limestone powder substitution level was also confirmed by Celik et al. [91]. The decrease in the permeability of cement matrix with the addition of rock dust is generally related to the filler effect, i.e., physical rock dust interaction. In addition, fine particles of rock dust block the continuity of capillary pores, which leads to the reduction of the capillary rise of water as well as permeability [29,91,140]. Dobiszewska et al. [56] observed the phenomenon of heteronucleation on the surface of rock dust particles. This phenomenon increases the production of crystallization nuclei, which leads to the densification of the cement paste and has a significant impact on the permeability reduction of the cement matrix when rock dust is added. The addition of rock powder accelerates cement hydration. It can be argued that hydrates fill free space between cement and dust particles, which directly contributes to the capillary pore content reduction and breaking of its continuity.

As mentioned earlier, the water absorption of concrete affects concrete durability. The ability of water absorption depends mainly on the distribution, size, shape, and tortuosity of pores, as well as their continuity [52]. Studies conducted by Almeida et al. [29] as well as and Celik and Marar [91] confirm that adding powdered limestone as a partial replacement for fine aggregate reduces concrete absorption. This is a consequence of the reduction of the pore content and the disruption of their continuity. The beneficial effects of rock powder on reducing water absorption of concrete were also confirmed by Alyamac and Aydin [94], Gameiro et al. [45], and Ulubeyli et al. [139], where marble dust was used as a partial fine aggregate substitute. Hameed et al. [137] further observed that adding of marble beyond 15–20% resulted in an increased water requirement in the concrete mixture due to the very high-specific surface area of the marble waste. This finding strengthened the results of previous studies conducted by Tasdemir [141], Gesoglu et al. [35], as well as Tsivilis et al. [52] indicating that the addition of rock powder of larger specific surface area than cement particles results in a reduction of porosity. The consequence of this is a lower absorption of concrete and greater resistance to the aggressive media penetration. However, some studies where quarry rock dust additives were used as a fine aggregate replacement indicated an increase of water absorption when a higher percentage of aggregate were substituted, resulting in a higher level of pores [91,142,143].

Further, the dissolution of compounds or chemical reactions between concrete and substance constituents occurs due to a chemical attack [144]. The most destructive agents that caused concrete deterioration are chlorides. Chloride ions penetrate concrete and replace hydroxide ions in cement hydrates during leaching. This leads to a lower pH of pore solution and, as a consequence, to the gradual disintegration of cement matrix. The resistance of concrete to the penetration of chloride ions is closely related to the concrete permeability and porosity. The ability of ion diffusion depends significantly on pore structure, the content of gel, and capillary pores. The effective diffusion coefficient decreases with the increase of gel pore contents and the disruption of capillary pore continuity [118]. As indicated earlier, heteronucleation on the rock dust particles surface leads to the increase in C-S-H phase content, and as a result to the densification of the cement matrix and change in pore size and structure [56]. The increase of fine pores content and break in continuity of capillary pores with the increase in C-S-H phase content is also observed. Thus, it results in a reduction of the rate of ion diffusion. The positive effect of limestone powder addition on the reduction of chloride ion permeability in concrete was noticed by Li et al. [145]. The enhancement in chloride resistance of concrete was observed also in the case of using granite powder as a partial replacement of fine aggregate [17]. In contrast, the conclusions made by Kepniak et al. [138] concerning the influence of the substitution for fine aggregate with limestone powder on concrete resistance to chloride corrosion observed an increase in the chloride ion concentration, and at the same time a reduction of total porosity with the increase of limestone powder addition. This indicates the faster chloride ion penetration which was confirmed by determination of the effective diffusion coefficient of chloride ions. Menadi et al. [22] have come to similar conclusions where the resistance to chloride ion penetration and gas permeability of concrete decrease with limestone powder increase, whereas water permeability is reduced. A negative effect of the influence of granite powder on chloride resistance of concrete was also observed by Vijayalakshmi et al. [54]. The presented results show that the concrete chloride permeability is proportional to the substitution rate, and the penetration rate increases with an increase in granite powder share. Vijayalakshmi et al. [54] stated that increase in the permeability of chloride ions is attributed to poor compaction, which results in higher porosity and a discontinuous pore system. This leads also to an increase in the carbonation depth value of the concrete with the increase in granite powder waste substitution.

Kępniak et al. [138] noticed an increase in the sulfate attack resistance of concrete with an increase in limestone powder amount, despite stated lower chloride resistance, as mentioned earlier. The authors noticed that, with an increase in limestone powder substitution level, the capillary pores content increases, in spite of the total porosity reduction. This favors the increase in the rate of chloride ion diffusion in concrete. The effect of a faster filling of smaller capillary pores with corrosion products prevents the further migration of sulphate ions from the solution, which results in the inhibition of the sulphate degradation process. The improvement of the mortar sulphate resistance as an effect of the incorporation of limestone powder was confirmed by Li et al. [145]. The decrease in sulphate resistance of the concrete with granite powder addition was noticed by Vijayalakshmi et al. [54]. This was caused by the contamination of granite powder with kerosene, diesel, and wax, which has been used during the process of sawing and polishing granite rock. In a study conducted by Inlangovana et al. [142], it was found that using quarry rock dust as fine aggregate increased concrete durability when compared to conventional concrete exposed to sulfate and acid action. As is known, the durability of concrete is directly related to the void structure and permeability of the concrete. Studies generally mention that more impermeable concrete can be produced thanks to the gap-filling effect of stone dust, but it is also seen that the materials used as fillers plays a much more effective role if it is finer-grained than cement, even if they are used as a fine aggregate substitution.

## 6. Conclusions

Concrete production is associated with environmental concerns since it consumes large amounts of raw materials, energy, and labor. Thus, worldwide there is an urgent demand to use by-products in building material production. In addition to materials that can be used as aggregate in the production of mortar and concrete, materials that can be substituted with cement clinker are also sought. The potential alternative materials that can be used in cement composites production as fine aggregate substitution include rock dust of different geological origin. The management of this waste is currently a serious problem for producers of mineral aggregates, asphalt mixture plants, and dimension stone industry. This indicates that more research concerning the management and utilization of these waste products in cement composites production is required. The review of past studies in this area synthesized in this manuscript provided the following valuable conclusions that can be considered in further studies.

The addition of rock powder significantly affects fresh concrete and mortar properties. The substitution for fine aggregate with rock dust leads generally to a significant decrease in workability. The much greater specific surface area of rock dust compared to fine aggregate results in a significant increase in water required by wet the particle surfaces, and thus poor workability. The solution to this problem is to use high water reducing admixtures to improve the workability of concrete. Therefore, there is a need to conduct research concerning the analysis of the influence of admixtures on concrete workability when the rock dust is used for fine aggregate substitution. As rock powder is very fine material, its addition leads to a reduction in bleeding and segregation. This is mainly the result of mix cohesion improvement by fine particles of rock dust and water retention enhancement.Improved mechanical properties of cement composites are due to the use of rock powder as a partial replacement for fine aggregate. The most important and dominant mechanism of beneficial rock dust interaction is connected with the filler effect, i.e., physical interaction. The space between the cement and aggregate grains is filled with very small particles of stone powder, which results in reducing the cement matrix porosity. With the addition of stone dust, the number of large capillary pores decreases and the content of small pores increases, which leads to sealing in the microstructure of the hardened cement paste and, accordingly, to a less permeable structure. As a result, cement composites with rock dust additive feature higher strength. Aside from the physical influence of stone dust on the cement matrix microstructure, other phenomena also occur. The rock dust grain surface is mainly the active center, which leads to the improvement of the properties and durability of cement composites from which heteronuclei of the C-S-H phase are formed. The heteronucleation on rock dust particles is much more favored by the fineness than geological origin of rock powder. As mentioned earlier, basalt dusts have some pozzolanic activity, which results in the increase of cement matrix density and thus strength improvement. In the case of using rock dust for fine aggregate substitution, the dominate role in property improvement is played by the filler effect, while the rock origin from which stone powder comes is less of importance. That is because analysed rock dust is in any case much finer than fine aggregate and possesses the greater specific surface area. The optimum fine aggregate replacement is about 20–30% and it depends more on rock dust fineness than its geological origin. With such a substitution level, an approximately 30% increase in mortar and concrete strength is observed.Reported results confirmed the positive effect of rock dust on concrete with an increase in the permeability and decrease in water absorption. Generally, the outcome is a result of the densification of cement matrix with fine rock powder particles, i.e., the filler effect. However, there were some contradictions regarding the influence of rock dust on permeability of concrete mainly to chloride ions. This depends on the finesses of the rock dust particles as compared to capillary pore and substitution level of fine aggregate with rock powder. In the case of sulphate attack, the addition of stone powder leads mainly to an improvement of sulphate resistance. Undoubtedly, further research is necessary to analyse effect of rock dust on cement composites durability, especially regarding chloride and sulphate corrosion, carbonation, and freeze-thaw resistance. Profound analysis concerning the influence of the fineness of rock dust on the penetration of chloride and sulphate ions is needed.Rock dust utilization in cement composite production requires the development of concrete design methods that allow to determine the optimal dust content in terms of obtaining the desired properties of both fluid concrete mix properties as well as hardened properties. Profound analysis is necessary to establish the optimum ratio for fine aggregate substitution with regard to the fineness of rock dust and addition of water reducing admixtures.Rock dust, which is currently considered as a by-product, can be used as a partial replacement for fine aggregates or even cement in cement mortars and concrete production. The utilization of rock dust waste is technically, economically, and ecologically justified and addresses the principle of sustainable development as it allows to reduce the consumption and dependency of natural resources for the production of cement composites and to manage the waste effectively.

As a result, considering the extensive studies in the literature, it can be concluded that rock dust is an environmentally friendly material that contributes economically to the mixture of cement-based materials. The use of rock dust for fine aggregate replacement at a certain amount in cement-based composites improves many fresh and hardened state properties. Therefore, rock dust should be taken into account in the optimum mix design of cement-based composites. Most of the studies in the literature also mentioned that, in addition to improving the properties of concretes, using stone dust in concrete led to the consumption of by-products, thus providing a twofold benefit.

When the results are evaluated for future studies, it is recommended that more research should be conducted on evaluating the usage of rock dust in high-performance concrete production and self-compacting concrete production, besides reactive powder concrete. The use of rock dust in the production process of cement composites requires the development of concrete design methods that allow the determination of the optimal rock dust content in terms of obtaining the desired properties of both the concrete mix and hardened concrete. If a careful analysis of the literature is performed, another important issue comes to the fore for future studies. Generally, the particle size distribution within the stone dust itself has not been taken into account by researchers. As known, it can be encountered in some cases that the particle size distribution in some intervals forms a significant part of the heap compared to other grain intervals. This situation directly affects many important parameters, such as water requirement, workability, gap-filling ability, etc., in concrete containing stone dust. For this reason, specifying the particle size distribution of these powder materials in studies instead of just calling them a material under 150 microns is recommended for future studies. A detailed analysis of particles size distribution can help to better interpret the results of the use of stone powder, as this affects the internal structure and many related properties.

## Figures and Tables

**Figure 1 materials-15-02947-f001:**
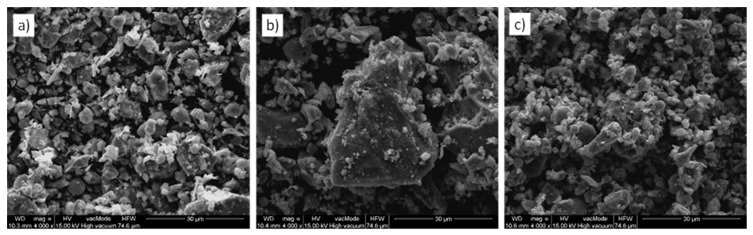
Microscopic images of rock dust: (**a**) limestone, (**b**) marble, (**c**) basalt.

**Figure 2 materials-15-02947-f002:**
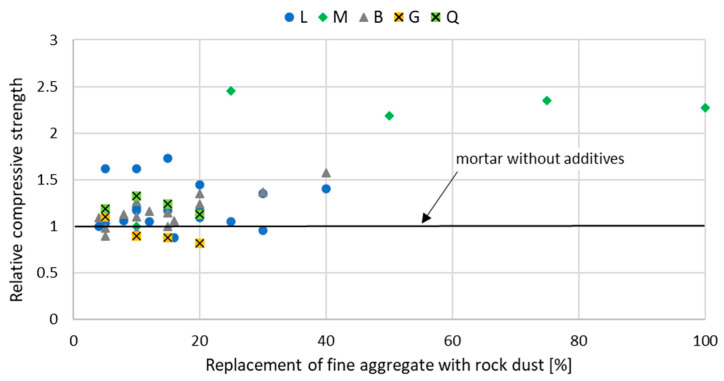
The effect of replacing fine aggregate with rock dust on the compressive strength of mortar after 7 days of curing: limestone dust (L) [46,55,95,97], marble dust (M) [67,76], basalt dust (B) [46,55,57,96], granite dust (G) [95], quartz dust (Q) [95].

**Figure 3 materials-15-02947-f003:**
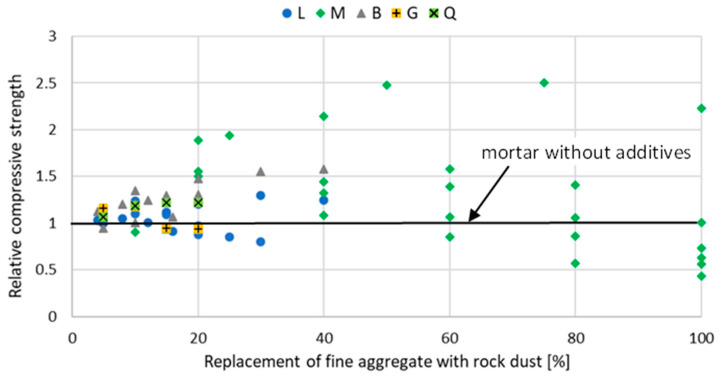
The effect of replacing fine aggregate with rock dust on the compressive strength of mortar after 28 days of curing: limestone dust (L) [46,55,95,97], marble dust (M) [63,67,76,86], basalt dust (B) [46,55,57,96], granite dust (G) [95], quartz dust (Q) [95].

**Figure 4 materials-15-02947-f004:**
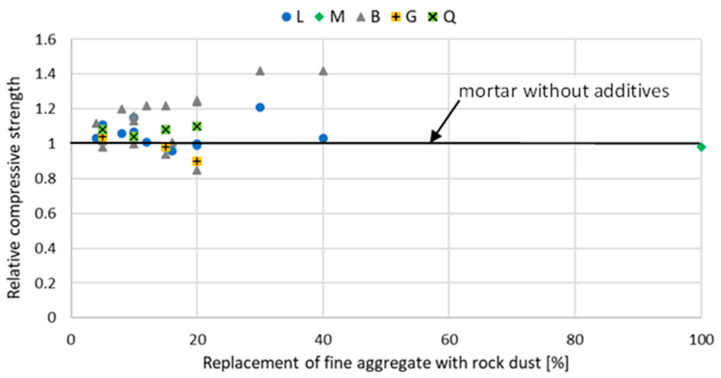
The effect of replacing fine aggregate with rock dust on the compressive strength of mortar after 90 days of curing: limestone dust (L) [46,55,95], marble dust (M) [63], basalt dust (B) [46,55,57,96], granite dust (G) [95], quartz dust (Q) [95].

**Figure 5 materials-15-02947-f005:**
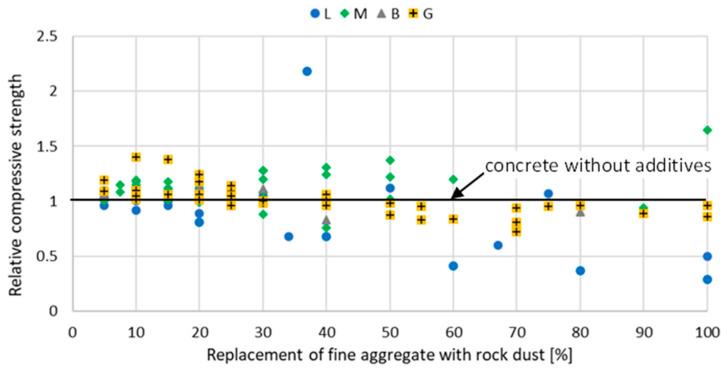
The effect of replacing fine aggregate with rock dust on the compressive strength of concrete after 7 days of curing: limestone dust (L) [13,33,91,92,98,99], marble dust (M) [12,30,33,34,61,93,94,100,101,102,103], basalt dust (B) [56,104,105], granite dust (G) [54,106,107,108,109,110].

**Figure 6 materials-15-02947-f006:**
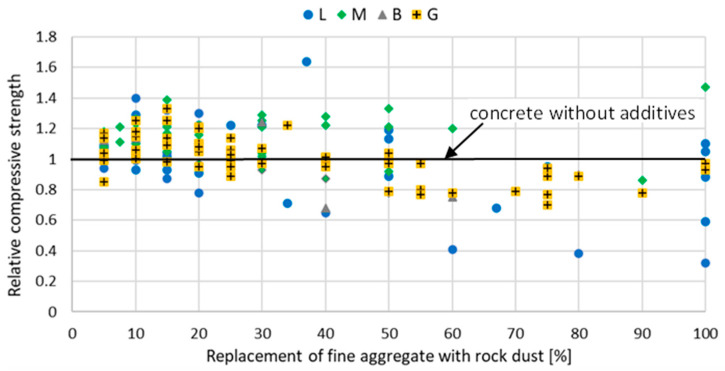
The effect of replacing fine aggregate with rock dust on the compressive strength of concrete after 28 days of curing: limestone dust (L) [13,22,33,44,91,92,98,99,111], marble dust (M) [12,30,33,34,61,93,94,100,101,102,103], basalt dust (B) [56,104,105], granite dust (G) [14,17,18,54,106,107,108,109,110].

**Figure 7 materials-15-02947-f007:**
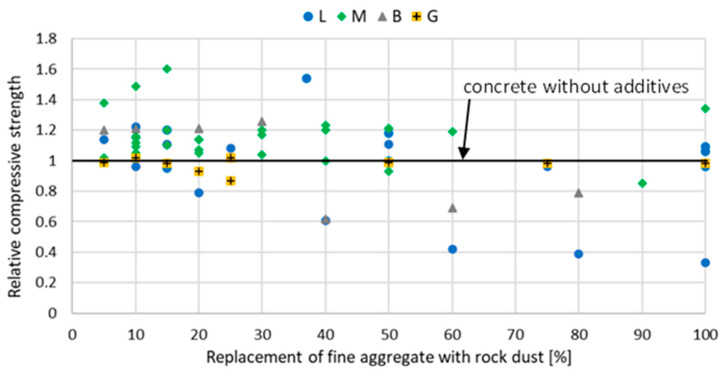
The effect of replacing fine aggregate with rock dust on the compressive strength of concrete after 90 days of curing: limestone dust (L) [22,33,92,98,99,111], marble dust (M) [12,33,34,93,94,100,102], basalt dust (B) [56,104,105], granite dust (G) [17,54,109].

**Table 1 materials-15-02947-t001:** Chemical composition of limestone [22,35,44,46,51,52], marble [30,35,38,48,53], granite [14,54], and basalt dust [46,48,55,56,57].

Oxide Composition	Limestone Dust	Marble Dust	Granite Dust	Basalt Dust
	[%]			
SiO_2_	0.22–12.90	0.18–6.01	51.98–85.50	44.59–56.33
CaO	42.30–56.09	40.73–83.22	1.82–5.90	6.42–12.80
Al_2_O_3_	0.18–2.70	0.29–0.73	2.10–16.30	5.76–20.70
Fe_2_O_3_	0.11–2.00	0.05–0.80	0.40–27.89	4.14–17.73
MgO	0.20–9.64	0.23–15.21	0.58–2.50	2.99–8.73
Na_2_O	0.01–0.54	0.06–2.44	2.02–3.69	0.84–4.11
K_2_O	0.03–0.60	0.05–1.80	2.99–4.12	0.35–1.62
SO_3_	0.01–0.88	0.08–0.56	0.05–1.80	0.02–1.10

## Data Availability

The study did not report any data.

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
