# Peer review of "Influence of Rock Dust Additives as Fine Aggregate Replacement on Properties of Cement Composites—A Review"

_materials, 2022, doi:10.3390/ma15082947_

Round 1

Reviewer 1 Report

Authors have done very good work on "Influence of rock dust additives as fine aggregate replacement on cement composites properties – a review". I suggest authors to modify the manuscript by considering the following suggestions/comments mentioned. 1. Title: Suggest to change the title. I feel the amount of work done is not gelling with the title. Ex: A) Influence of rock dust additives as fine aggregate replacement on properties of cement composites – A review. B) Influence of rock dust additives as fine aggregate replacement on properties of cement concrete – a review. 2. I suggest authors to incorporate the comparison table for the properties (Both physical and chemical) of materials (Ex: Rock dust) 3. Please use either word microscopic or macroscopic, dont confuse the reader. 4. At the end after reviewing the articles published, I suggest authors to write the overall discussion about each property reviewed/studied. 5. Conclusions are very general after reviewing more than 100 papers, suggest to strengthen the conclusions part. 6. Incorporate Perspectives and suggestion for future researches at the end after conclusion. This will catch the readers, researchers towards meaningful research and helps to get more citations as well. 7. I suggest authors to cite at least 8-10 articles (Recently published) from Materials Journal.

Author Response

Reviewer 1

  1. Title: Suggest to change the title. I feel the amount of work done is not gelling with the title. Ex: A) Influence of rock dust additives as fine aggregate replacement on properties of cement composites – A review. B) Influence of rock dust additives as fine aggregate replacement on properties of cement concrete – a review.

Response to comment 1

According to the reviewer’s comment the title seen below has been selected as new title of the article.

“Influence of rock dust additives as fine aggregate replacement on properties of cement composites – A review”

  1. I suggest authors to incorporate the comparison table for the properties (Both physical and chemical) of materials (Ex: Rock dust)

Response to comment 2

A table titled “Table 1. Chemical composition of limestone [22,35,44,46,51,52], marble [30,35,38,48,53], granite [14,54] and basalt dust [46,48,55-57]” including the properties of materials has been added to the article.

  1. Please use either word microscopic or macroscopic, dont confuse the reader.

Response to comment 3

The word macroscopic is only written in Figure 1. This is a typo, and it has been corrected according the reviewer’s comment

  1. At the end after reviewing the articles published, I suggest authors to write the overall discussion about each property reviewed/studied.

Response to comment 4

According to the reviewer’s comment, the overall discussion about each property has been added to the related parts.

  1. Conclusions are very general after reviewing more than 100 papers, suggest to strengthen the conclusions part.

Response to comment 5

According to the suggestion, the conclusion part of the study has been improved.

  1. Incorporate Perspectives and suggestion for future researches at the end after conclusion. This will catch the readers, researchers towards meaningful research and helps to get more citations as well.

Response to comment 6

A suggestion has been added to the end of conclusion part. We would like to thank to reviewer for this important suggestion.

  1. I suggest authors to cite at least 8-10 articles (Recently published) from Materials Journal.

Response to comment 7

According to the reviewer`s suggestion, some recently published articles have been added.

Authors would like to thank the Reviewer for all very valuable comments.

Reviewer 2 Report

In this paper, the authors studied the Influence of rock dust additives as fine aggregate replacement on cement composites properties. the paper is interesting and useful, some minor comments are needed:

-The introduction is well written, but  found some old Refs, so I encourage the authors to update some Refs and add new Refs (2022 and 2021)

-In section 3, I suggest to the authors to add one paragraph about the utilization of the concrete for radiation shielding. since, recently many new researches reported that the concrete is a suitable material in radiation shielding applications

-the conclusion is long, please shorten it

-please enhance the resolution of the figures

Author Response

Reviewer 2

The introduction is well written, but found some old Refs, so I encourage the authors to update some Refs and add new Refs (2022 and 2021)

Response to the reviewer:

According to reviewer’s comment, some newly published articles have been added to the article.

In section 3, I suggest to the authors to add one paragraph about the utilization of the concrete for radiation shielding. since, recently many new researches reported that the concrete is a suitable material in radiation shielding applications

Response to the reviewer:

This is very important comment but the article is only focused on the fresh state and hardened state properties of cementitious materials. If we try to study on radiation shielding, it may be very difficult to compare all rock dust discussed in this article because radiation shielding is directly related to the unit weight of concrete. 

The conclusion is long, please shorten it

Response to the reviewer:

In our opinion, shortening this part may be more good but the authors could not decide for this suggestion because other two reviewers advised to extend and strengthen this part. Considering that this article is a state-of-art article, the authors have added some additional comments to this part.

Please enhance the resolution of the figures

Response to the reviewer:

According to reviewer’s comment, the resolution of all figures have been improved. We would like to thank the review for this valuable comment.

Authors would like to thank the Reviewer for all very valuable comments.

Reviewer 3 Report

Overall, the work is interesting as it is focused on the utilization of waste dust powders in concrete production, which can assist both the concrete and stone industries. However, it needs significant improvements. Other properties of concrete incorporating rock dust should be analyzed and more explanatory figures need to be added. Figures need further adjustment in order to make them more readable (decreasing the size of points may help). The conclusion section needs major revision; it needs to be focused on the purpose of the paper and provide the reader with brief but comprehensive conclusions. Further recommendations can be found below:

Lines 30-31: “In this assessment and review concrete and mortar mixture properties, mechanical properties and durability were considered.”

Please clarify what you mean by mixture properties.

Line 107: section 2. Rock dust characteristics

There are several parts in this section that can be moved to the introduction. As of the title of this section, it is expected to see the discussion revolving around the characteristics of rock dust.  

Line 203-205: this section is a bit unclear; what does “their properties” refer to?

Line 206: please clarify what you mean by “mixture properties”. Is it fresh properties?

Lines 427-428: “it should be noticed that the rock dust specific surface area affects to a greater extent the mortar and concrete mechanical properties when rock dust is used as a partial cement substitution rather than as fine aggregate replacement.” This statement is not clear. In most cases, when cement is replaced with an inert material, the optimum ratio is reported to be around 10%. So, even if 10% of fine aggregates is replaced with dust, there should be the same effect from the contribution of powder grains.

Lines 471: “with a certain rock dust content replacing fine aggregate, the mortar and concrete strengths decrease were observed  “ is not clear.

Author Response

Reviewer 3

Overall, the work is interesting as it is focused on the utilization of waste dust powders in concrete production, which can assist both the concrete and stone industries. However, it needs significant improvements. Other properties of concrete incorporating rock dust should be analyzed and more explanatory figures need to be added. Figures need further adjustment in order to make them more readable (decreasing the size of points may help). The conclusion section needs major revision; it needs to be focused on the purpose of the paper and provide the reader with brief but comprehensive conclusions. Further recommendations can be found below:

Response to the reviewer:

The authors found only single articles concerning the analysis of the influence of waste rock dust on other concrete properties like for example permeability, water absorption, porosity, freeze resistance. It is difficult to draw some comprehensive conclusions from these a few articles. Therefore, authors added such a conclusion regarding the necessity to perform studies on the analysis of these properties to the summary of the article: “The suitability of concrete for use in construction is also evidenced by its other properties, such as permeability, water absorption, porosity, freeze resistance. There are lack of comprehensive research concerning the analysis of the influence of waste rock dust on such a concrete properties. Therefore, further research should be carried out on the influence of rock dust on these properties of concrete and mortar.”

Lines 30-31: “In this assessment and review concrete and mortar mixture properties, mechanical properties and durability were considered.”

Please clarify what you mean by mixture properties.

Response to the reviewer:

Authors mean fresh concrete and mortar properties, i.e., workability, segregation, and bleeding. According to reviewer`s suggestion it was added to the article.

 Line 107: section 2. Rock dust characteristics

There are several parts in this section that can be moved to the introduction. As of the title of this section, it is expected to see the discussion revolving around the characteristics of rock dust.  

Response to the reviewer:

If it is not considered mandatory by the reviewer, the authors suggest not to move this section to the introduction of the article.

Line 203-205: this section is a bit unclear; what does “their properties” refer to?

Response to the reviewer:

“Their properties” refers to concrete and mortar properties. The sentence has been changed to clearer:

“Thus, it should be stated that considering the applicable standards, the addition of dusts as mineral fillers in the composition of mortars and concretes is determined by these cement composites properties, which may not be affected by the addition of dusts.”

Line 206: please clarify what you mean by “mixture properties”. Is it fresh properties?

Response to the reviewer:

Authors mean properties of fresh concrete and mortar. The title of this section has been changed: “Fresh concrete and mortar properties”.

Lines 427-428: “it should be noticed that the rock dust specific surface area affects to a greater extent the mortar and concrete mechanical properties when rock dust is used as a partial cement substitution rather than as fine aggregate replacement.” This statement is not clear. In most cases, when cement is replaced with an inert material, the optimum ratio is reported to be around 10%. So, even if 10% of fine aggregates is replaced with dust, there should be the same effect from the contribution of powder grains.

Response to the reviewer:

The fine aggregate specific surface area is much smaller than that of rock dust. Therefore, the substitution of sand by rock dust leads to increase in specific surface area of particles in any case and it does not depend on the finesses of rock dust. In the case of replacing cement by rock dust the specific of the surface area of inert materials is of much greater importance. If the rock dust if finer, then cement the introduction of this inert additive leads to increase in overall specific surface area of granular material which results in densification of the cement matrix.

Lines 471: “with a certain rock dust content replacing fine aggregate, the mortar and concrete strengths decrease were observed” is not clear.

Response to the reviewer:

The general conclusion drawn from the most reviewed articles is that the addition of rock dust replacing fine aggregate leads to mortar and concrete strength improvement. However, a few researchers observed decrease in mortar and concrete strength with fine aggregate substitution by rock dust.

Authors would like to thank the Reviewer for all very valuable comments.

Round 2

Reviewer 1 Report

The authors have modified the manuscript by considering all the suggestions, It can be considered now for publication. Best wishes to all the authors to work more on cement and materials for better and sustinable environment. 

Author Response

The authors have modified the manuscript by considering all the suggestions, It can be considered now for publication. Best wishes to all the authors to work more on cement and materials for better and sustinable environment.

Authors would like to thank the Reviewer very much for all very valuable comments.

Reviewer 3 Report

Table 1 shows that the target of the present review is to study the effect of waste dust from marble, granite, limestone, and basalt industries as a partial replacement of fine aggregates in concrete. The literature has widely investigated the properties of concrete and mortar containing these waste materials. However, the review of the durability properties in this work is still limited. In addition, in several parts, the justifications and results are from studies that used waste powders as cement replacement, which might be confusing for the reader as the target here is to evaluate the role of waste powders as fine aggregate replacement. For example, line 556: “The addition of limestone dust of particles larger than cement grains leads to increase concrete permeability. Whereas the substitution of cement with basalt and marble dust of finer grains results in decrease of permeability. Celik et al. [91] also observed greater resistance to chloride ions migration of concrete with basalt dust additive of fineness similar to cement.” So, it is not clear what would be the durability properties if these waste materials partially replace fine aggregates. According to this, it is also not clear that what is mentioned in line 568: “but it is also seen that the material used as filler plays a much more effective role if it is finer-grained than cement.” is based on the studies that used the powders as cement replacement or fine aggregate replacement.

Line 610: “The use of rock dust depending on its type and particle size at a certain amount in cement-based composites improves many fresh and hardened state properties” please clarify if this represents the case of fine aggregate replacement?

The conclusions still need significant improvements. There are no quantitative conclusions. What is the optimum replacement ratio? What are the factors influencing these optimum ratios? How does stone dust influence each concrete property?

Line 596: What about the pozzolanic activity of basalt, as mentioned in line 427?

Lines 593-596: “Aside from the physical influence of stone dust on the cement matrix microstructure, other phenomena also occur. The rock dust grain surface is mainly the active center, which leads to the improvement of the properties and durability of cement composites from which heteronuclei of the C-S-H phase are formed.” Please clarify if this can be applied to all types of waste powders, considering the text in lines 556-558.

Table1, if the numbers in this table show percentage, then the comma (,) needs to be replaced with dot (.) to present the decimals.

Figure 1: the SEM of granite is missing. Also, the source of other SEMs is not specified.

Line 300: “the use of fillers affects the workability negatively by effectively changing the water/cement in the concrete. water/cement in the concrete.” Please modify if you mean water to binder ratio as waste powders studied in this work do not possess cementitious properties.

Author Response

Table 1 shows that the target of the present review is to study the effect of waste dust from marble, granite, limestone, and basalt industries as a partial replacement of fine aggregates in concrete. The literature has widely investigated the properties of concrete and mortar containing these waste materials. However, the review of the durability properties in this work is still limited. In addition, in several parts, the justifications and results are from studies that used waste powders as cement replacement, which might be confusing for the reader as the target here is to evaluate the role of waste powders as fine aggregate replacement. For example, line 556: “The addition of limestone dust of particles larger than cement grains leads to increase concrete permeability. Whereas the substitution of cement with basalt and marble dust of finer grains results in decrease of permeability. Celik et al. [91] also observed greater resistance to chloride ions migration of concrete with basalt dust additive of fineness similar to cement.” So, it is not clear what would be the durability properties if these waste materials partially replace fine aggregates. According to this, it is also not clear that what is mentioned in line 568: “but it is also seen that the material used as filler plays a much more effective role if it is finer-grained than cement.” is based on the studies that used the powders as cement replacement or fine aggregate replacement.

Response to the reviewer:

Authors would like to thank you very much for this comment. Unfortunately, only limited, or few articles published concern on assessing the influence of fine aggregate substitution with rock dust on mortar and concrete durability which indicates the need for further research. Authors found couple articles and sections pertaining to this topic were greatly modified in accordance with the Reviewer suggestion. Also, the citation of article concerning the use of rock dust as a cement replacement was a blunder on the authors’ side.

  1. Concrete and mortar durability

Galetakis and Soultana [39], as well as many other studies, asserted that permeability is one of the most important factors characterized the durability of concrete. The permeability of concrete is often measured based on its resistance to allow penetration and movement of aggressive substances within its mass. The published research results indicate that concrete with addition of different mineralogical origin rock waste demonstrated lower water permeability as compared with conventional concrete [33,35,92,137,138]. A study on the use of marble waste as coarse aggregate replacement conducted by Ulubeyli et al. [139] found that marble waste acted as filler reducing the gaps within the hardened concrete, thus, provides a lower porous structure of concrete. It can be stated that water permeability depends primarily on the capillary pores volume. However, Kurdowski [118] concluded that permeability is determined not only to the total porosity but also to the distribution, tortuosity, shape of pores as well as its size and continuity. The study conducted by Holly et al. [140] supported this concept by demonstrating a remarkable impact of the interconnectivity of cement paste pores and the pore size distribution on permeability. Menadi [22] observed reduction in the water penetration depth with increase in limestone powder content. This is the effect of improvement of pore structure in interfacial transition zone. The increase in concrete water permeability with increase in limestone powder substitution level was also confirmed by Celik et al. [91]. Decrease in the permeability of cement matrix with addition of rock dust is generally related to the filler effect, i.e. physical rock dust interaction. In addition, fine particles of rock dust block the capillary pores continuity which leads to the reduction of capillary rise of water as well as permeability [29,91,140]. Dobiszewska et al. [56] observed the phenomenon of heteronucleation on the surface of rock dust particles. This phenomenon increases the production of crystallization nuclei which leads to the densification of the cement paste and has a significant impact on the permeability reduction of the cement matrix when rock dust is added. The addition of rock powder accelerates cement hydration. It can be argued that hydrates fill free space between cement and dust particles which directly contributes to the capillary pore content reduction and breaking of its continuity.

As mentioned earlier, water absorption of concrete affects concrete durability. The ability of water absorption depends mainly on distribution, size, shape, tortuosity of pores and their continuity [52]. Studies conducted by Almeida et al. [29], and Celik and Marar [91] confirm that adding powdered limestone as a partial replacement for fine aggregate reduces concrete absorption. This is a consequence of the reduction of the pore content and the disruption of their continuity. The beneficial effects of rock powder on reducing water absorption of concrete were also confirmed by Alyamac and Aydin [94], Gameiro et al. [45] and Ulubeyli et al. [139] where marble dust was used as partial fine aggregate substitute. Hameed et al. [137] further observed that adding of marble beyond 15 to 20% resulted in an increased water requirement in the concrete mixture due to very high-specific surface area of the marble waste. This finding strengthened previous studies results conducted by Tasdemir [141], Gesoglu et al. [35] as well as Tsivilis et al. [52] that the addition of rock powder of larger specific surface area than cement particles results in reduction of porosity. The consequence of this is lower absorption of concrete and greater resistance to the aggressive media penetration. However, some studies where quarry rock dust additives were used as fine aggregates replacement indicated an increase of water absorption when a higher percentage of aggregate were substituted resulting to higher level of pores [91,142,143].

Further, dissolution of compounds or chemical reactions between concrete and substances constituents occurs due to a chemical attack [144]. The most destructive agents that caused concrete deterioration are chlorides. Chloride ions penetrate concrete and replace hydroxide ions in cement hydrates during leaching. This leads to lower pH of pore solution and as a consequence to the gradual disintegration of cement matrix. Resistance of concrete to the penetration of chloride ions is closely related to the concrete permeability and porosity. Ability of ions diffusion depends significantly on pore structure, the content of gel and capillary pores. The effective diffusion coefficient decreases with gel pores content increase and disruption of capillary pores continuity [118]. As it was indicated earlier, heteronucleation on the rock dust particles surface leads to the increase in C-S-H phase content and as a result to densification of cement matrix, and change pore size and structure [56]. The increase of fine pores content and break in continuity of capillary pores with the increase in C-S-H phase content is also observed. Thus, it results in the rate of ions diffusion reduction. The positive effect of limestone powder addition on reduction of chloride ion permeability in concrete was noticed by Li et al. [145]. The enhancement in chloride resistance of concrete was observed also in the case of using granite powder as a partial replacement of fine aggregate [17]. In contrary, the conclusions made by Kepniak et al [138] concerning the influence of substitution for fine aggregate with limestone powder on concrete resistance to chloride corrosion observed increase in the chloride ions concentration, and at the same time a reduction of total porosity with increase of limestone powder addition. This indicates the faster chloride ion penetration which was confirmed by determination of effective diffusion coefficient of chloride ions. Menadi et al. [22] have come to similar conclusions where resistance to chloride ions penetration and gas permeability of concrete decrease with limestone powder increase whereas water permeability is reduced. Negative effect of the influence of granite powder on chloride resistance of concrete was also observed by Vijayalakshmi et al. [54]. The presented results show that the concrete chloride permeability is proportional to the substitution rate, and the penetration rate increases with increase in granite powder share. Vijayalakshmi et al. [54] stated that increase in the permeability of chloride ions is attributed to poor compaction which result in higher porosity and a discontinuous pore system. This leads also to increase in carbonation depth value of the concrete with the increase in granite powder waste substitution.

Kępniak et al. [138] noticed the increase of sulfate attack resistance of concrete with the increase in limestone powder amount, despite stated lower chloride resistance, as mentioned earlier. Authors noticed that with increase in limestone powder substitution level, the capillary pores content increases, in spite of total porosity reduction. This favors the increase in rate of chloride ions diffusion in concrete. The effect of faster filling of smaller capillary pores with corrosion products, prevents the further migration of sulphate ions from the solution which results in inhibition of sulphate degradation process. The improvement of the mortar sulphate resistance as an effect of the incorporation of limestone powder was confirmed by Li et al. [145]. The decrease in sulphate resistance of concrete with granite powder addition was noticed by Vijayalakshmi et al. [54]. This was caused by contamination of granite powder with kerosene, diesel and wax which has been used during the process of sawing and polishing of granite rock. In a study conducted by Inlangovana et al. [142], it was found that using quarry rock dust as fine aggregate increased concrete durability when compared to conventional concrete exposed to sulfate and acid action. As it is known, the durability of concrete is directly related to the void structure and permeability of the concrete. Studies generally mention that more impermeable concrete can be produced thanks to the gap-filling effect of stone dust, but it is also seen that the material used as filler plays a much more effective role if it is finer-grained than cement even if they are used as fine aggregate substitution.

Line 610: “The use of rock dust depending on its type and particle size at a certain amount in cement-based composites improves many fresh and hardened state properties” please clarify if this represents the case of fine aggregate replacement?

Response to the reviewer:

This statement represents the case of fine aggregate replacement. It was written more clearly in the article:

“The use of rock dust for fine aggregate replacement at a certain amount in cement-based composites improves many fresh and hardened state properties.”

The conclusions still need significant improvements. There are no quantitative conclusions. What is the optimum replacement ratio? What are the factors influencing these optimum ratios? How does stone dust influence each concrete property?

Response to the reviewer:

Section 5. Conclusions has been modified and improved according to Reviewer suggestion.

Concrete production is associated with environmental concerns since it consumes large amount of raw materials, energy, and labor. Thus, worldwide there is an urgent demand to use by-products in building materials production. In addition to materials that can be used as aggregate in the production of mortar and concrete, materials that can be substituted with cement clinker are also sought. The potential alternative materials that can be used in cement composites production as fine aggregate substitution is rock dust of different geological origin. The management of this waste is currently a serious problem for producers of mineral aggregates, asphalt mixture plants, and dimension stone industry. This indicates that more research into the management and utilization of these waste products in cement composites production is required. The review of past studies in this area synthesized in this manuscript provided the following valuable conclusions that can be considered in further studies.

  1. The addition of rock powder significantly affects fresh concrete and mortar properties. The substitution for fine aggregate with rock dust leads generally to significant decrease in workability. The much greater specific surface area of rock dust compared to fine aggregate, results in significant increase in water requirement to wet the particles surface and thus poor workability. The solution of this problem is to use high water reducing admixtures to improve workability of concrete. Therefore, there is a need to conduct research concerning analysis of influence of admixtures on concrete workability when the rock dust is used for fine aggregate substitution. As rock powder is very fine material its addition leads to a reduction in bleeding and segregation. This is mainly the result of mix cohesion improvement by fine particles of rock dust and water retention enhancement.
  2. Improved mechanical properties of cement composites are due to the use of rock powder as a partial replacement for fine aggregate. The most important and dominant mechanism of beneficial rock dust interaction is connected with the filler effect, i.e. physical interaction. The space between the cement and aggregate grains is filled with very small particles of stone powder, which results in reducing the cement matrix porosity. With the addition of stone dust, the large capillary pores number decreases, and the small pores content increases which leads to sealing in the microstructure of the hardened cement paste and, accordingly, to a less permeable structure. As a result, cement composites with rock dust additive feature higher strength. Aside from the physical influence of stone dust on the cement matrix microstructure, other phenomena also occur. The rock dust grain surface is mainly the active center, which leads to the improvement of the properties and durability of cement composites from which heteronuclei of the C-S-H phase are formed. The heteronucleation on rock dust particles is much more favored by the fineness than geological origin of rock powder. As it was mentioned earlier basalt dusts have some pozzolanic activity, which results in the increase of cement matrix density and thus strength improvement. In the case of using rock dust for fine aggregate substitution, the dominate role in properties improvement is played by filler effect, and the rock origin from which stone powder comes, is less of importance. That is because analysed rock dust is in any case much finer than fine aggregate and possess the greater specific surface area. The optimum fine aggregate replacement is about 20-30% and it mostly depend on rock dust fineness than its geological origin. With such substitution level about 30% increase in mortar and concrete strength is observed.
  3. Reported results confirmed the positive effect of rock dust on concrete with increase permeability and decrease in water absorption. Generally, the outcome is a result of densification of cement matrix with fine rock powder particles, i.e., filler effect. However, there were some contradictions regarding the influence of rock dust on permeability of concrete mainly to chloride ion. This depends on the finesses of the rock dust particles as compared to capillary pore and substitution level of fine aggregate with rock powder. In the case of sulphate attack, the addition of stone powder leads mainly to improvement of sulphate resistance. Undoubtedly, further research is necessary to analyse effect of rock dust on cement composites durability, especially regarding chloride and sulphate corrosion, carbonation, and freeze-thaw resistance. The profound analysis concerns the influence of fineness of rock dust on penetration of chloride and sulphate ions is needed.
  4. Rock dust utilization in cement composites production requires the development of concrete design methods that allow to determine the optimum dust content in terms of obtaining the desired properties of both workable concrete mix properties as well as hardened properties. Profound analysis is necessary to establish the optimum ratio for fine aggregate substitution regard to the fineness of rock dust and addition of water reducing admixtures.
  5. Rock dust, which is currently considered as a by-product, can be used as a partial replacement for fine aggregates or even cement in cement mortar and concrete production. The utilization of rock dust waste is technically, economically, and ecologically justified and addresses the principle of sustainable development as it allows to reduce the consumption and dependency of natural resources for the production of cement composites and to manage the waste effectively.

As a result, considering the extensive studies reported in literature, it can be concluded that rock dust is an environmentally friendly material that contributes economically to the mixture of cement-based materials. The use of rock dust depending on its type and particle size at a certain amount in cement-based composites improves many fresh and hardened state properties. Therefore, rock dust should be taken into account in the optimum mix design of cement-based composites. Most of the studies in the literature also mentioned that in addition to improving the properties of concretes, using stone dust in concrete led to the consumption of by-products, thus providing a twofold benefit.

When the results are evaluated for future studies, it is recommended that more research should be conducted on evaluating the usage of rock dust in high-performance concrete production, self-compacting concrete production besides reactive powder concrete. The use of rock dust in the production process of cement composites requires the development of concrete design methods that allow the determination of the optimal rock dust content in terms of obtaining the desired properties of both the concrete mix and hardened concrete. If a careful analysis of the literature is made, another important issue comes to the fore for future studies. Generally, the particle size distribution within the stone dust itself has not been taken into account by researchers. As known, it can be encountered in some cases that the particle size distribution in some intervals forms a significant part of the heap compared to other grain intervals. This situation directly affects many important parameters such as water requirement, workability, gap-filling ability, etc. in concrete containing stone dust. For this reason, specifying the particle size distribution of these powder materials in studies instead of just calling them a material under 150 microns is recommended for future studies. Detailed analysis of particles size distribution can help to better interpret the results of the use of stone powder, which affects the internal structure and many related properties.

Line 596: What about the pozzolanic activity of basalt, as mentioned in line 427?

Response to the reviewer:

The pozzolanic activity is just now added to the conclusion according to Reviewer`s suggestion:

“As it was mentioned earlier basalt dusts have some pozzolanic activity, which results in the increase of cement matrix density and thus strength improvement.”

Lines 593-596: “Aside from the physical influence of stone dust on the cement matrix microstructure, other phenomena also occur. The rock dust grain surface is mainly the active center, which leads to the improvement of the properties and durability of cement composites from which heteronuclei of the C-S-H phase are formed.” Please clarify if this can be applied to all types of waste powders, considering the text in lines 556-558.

Response to the reviewer:

It has been changed in the conclusion.

Table1, if the numbers in this table show percentage, then the comma (,) needs to be replaced with dot (.) to present the decimals.

Response to the reviewer:

It has been changed in Table 1.

Figure 1: the SEM of granite is missing. Also, the source of other SEMs is not specified.

Response to the reviewer:

The SEM analysis has been done by one of the authors. Therefore, it is own source.

Line 300: “the use of fillers affects the workability negatively by effectively changing the water/cement in the concrete. water/cement in the concrete.” Please modify if you mean water to binder ratio as waste powders studied in this work do not possess cementitious properties.

Response to the reviewer:

As rock dust do not possess cementitious properties, we mean water/cement ratio.

Authors would like to thank the Reviewer very much for all very valuable comments.
